

# Synergistic effect of seaweed extract and boric acid and/or calcium chloride on productivity and physico-chemical properties of Valencia orange

Abdullah Alebidi and Mahmoud Abdel-Sattar

Department of Plant Production, College of Food and Agriculture Sciences, King Saud University, Riyadh, Saudi Arabia

## ABSTRACT

Many citrus species and cultivars are grown successfully in tropical and subtropical countries, as well as in arid and semi-arid regions with low levels of organic matter and low cation exchange, resulting in lower nutrient uptake by the plant. The essential nutrients needed for citrus flowering and fruit set are limited in winter due to a reduction in transpiration rate, negatively effecting vegetative growth, flowering, yield, and fruit quality. The present investigation was carried out to assess the nutritional status, fruit yield parameters, and fruit quality of Valencia orange trees after foliar spraying of seaweed extract (SW) combined with calcium chloride and boric acid and their combinations in the 2020/2021 and 2021/2022 seasons. The treatments were arranged in a split-plot design (three levels spraying seaweed extract × four levels spraying calcium chloride and boric acid and their combinations × four replicates × one tree/replicate). The results indicated that all of the characteristics measured, including leaf chlorophyll, leaf mineral contents, fruit yield parameters, fruit physical properties, and fruit chemical properties, were significantly affected by the foliar spraying of seaweed extract (SW) combined with calcium chloride and boric acid and their combinations. Although all treatments increased the productivity and the physical and chemical properties of Valencia orange fruits compared to the control, a treatment of 10 g/L SW combined with 0.5 g/L boric acid and 1 g/L calcium chloride produced superior results. This ratio of SW, boric acid, and calcium chloride is therefore recommended to enhance productivity and improve the physico-chemical properties of Valencia orange for greater fruit yield.

# INTRODUCTION

Citrus is an important fruit crop in tropical and subtropical countries, with unique characteristics such as a long maturation period, the ability to store fruits on the tree after maturity, and many nutritional value qualities (*Şekerli & Tuzcu, 2020*). Citrus fruits are one of the main economic fruit crops worldwide (*Domingues et al., 2021a*). Citrus fruits are also a healthy part of the human diet because of their high nutritional value

Corresponding author
Mahmoud Abdel-Sattar,
mmarzouk1@ksu.edu.sa

(*Liu, Heying & Tanumihardjo, 2012*; *Al-Saif et al., 2022a*), particularly their levels of vitamin C, which increases resistance to influenza and decreases calcium oxalate accumulation in the kidneys (*Haleblian et al., 2008*; *Alshallash et al., 2022*). Oranges, in both fruit and juice forms, are extremely popular throughout the world due to their flavor and nutritional value (*Bai et al., 2009*; *Pan et al., 2023*). Valencia (*Citrus sinensis* L. Osbeck) is considered one of the best and most popular late-maturing citrus cultivars with a high economic value; they are prized for their high productivity and good juice quality and bring higher prices in the export market in most citrus-producing countries (*Papadakis, Protopapadakis & Therios, 2008*; *Hervalejo et al., 2021*). Valencia's late-maturing fruit products are primarily consumed as juice and are in high demand globally (*Domingues et al., 2021b*). Valencia orange juice has the most carotenoids and the most complex pigment pattern of any sweet orange (*Zacarías-García et al., 2022*). Because of this, the Valencia orange is the most frequently planted citrus cultivar in the world.

Many citrus species and cultivars are effectively grown in arid and semi-arid regions such as the Riyadh region of Saudi Arabia. These regions have coarse-textured, highly calcareous soils that are poor in organic matter and have a low cation exchange, with high nutrient leaching losses resulting in lower nutrient uptake by the plant, affecting vegetative growth, flowering, yield, and fruit quality (*Salama, 2015*; *Al-Omran et al., 2020*). Fruit trees cultivated in such soils require extra care in fertilizer management strategies to preserve yield potential and desired fruit quality while lowering production costs. Therefore, alternate or complementary fertilization methods are needed. Foliar fertilization aims to minimize nutrient loss and maximize the nutrient use efficiency of the crops, and is considered one of the most effective factors in controlling growth, yield, and fruit quality of fruit crops. In addition, foliar application does not cause the serious environmental problems that other fertilization methods cause, which lead to pollution of soil and water and the accumulation of harmful residual substances in fruits, impacting human health (*Diacono & Montemurro, 2015*). Foliar fertilization is highly effective at delivering plant nutrients because nutrients reach the leaves and are absorbed equally, reducing the amount of fertilizer needed (*Medan, 2020*). Foliar sprays also provide nutrients to plants more quickly than soil applications. Micronutrient foliar spray is 7–21 times more effective than soil application (*Zaman et al., 2020*). Therefore, the foliar application of seaweed extract-based fertilizer, in combination with calcium and boron treatments, offers an easy, economical, safe, and promising approach to enhancing the yield, fruit quality, and nutritional value of Valencia oranges. This method facilitates accelerated plant growth, improves blooming and production, and enhances nutritional content, especially since calcium and boron are not very mobile in plants (*Bezerra et al., 2023*). This method also helps ensure the safety of fruits for local consumers, has a high exportation potential, and is economical (*Trautmann et al., 2014*; *Abdel-Aziz & El-Azazy, 2016*; *Yang et al., 2023*).

Seaweed extract is a natural and cheap source of organic matter and minerals since it contains macronutrients (*i.e.*, P, K, Ca, and Mg), micronutrients (*i.e.*, Cu, Fe, Mn, and Zn), some growth regulators (*i.e.*, auxins and cytokinins, gibberellins, and polyamines) proteins, vitamins, polysaccharides, sterols, polyphenols, antioxidants, pigments, and antimicrobial agents (*Spinelli et al., 2009*; *Aremu et al., 2016*; *Bulgari, Franzoni & Ferrante, 2019*;

Ali, Ramsubhag & Jayaraman, 2021; Al-Saif et al., 2023; Rana et al., 2023). Consequently, seaweed extract can improve plant growth, nutrient uptake, the photosynthesis process, nutritional status, vegetative growth, yield, fruit quality, and resistance of plants to unfavorable stresses (Bulgari, Franzoni & Ferrante, 2019; Ali, Ramsubhag & Jayaraman, 2021), especially under semi-arid and desert conditions (Prasad et al., 2010; Almaroai & Eissa, 2020). Seaweed extract can also lengthen the shelf life of produce (Norrie & Keathley, 2005; Gemida, Ardeña & Pillones, 2023). The foliar application of low concentrations of seaweed extract can induce a variety of positive physiological plant responses, including improved shoot and leaf growth, flower formation quality, nutritional status of fruit trees, growth, productivity, fruit quality attributes, fruit set percentage, and fruit size (Basak, 2008; Khan et al., 2009; Mosa et al., 2023; Rana et al., 2023).

Citrus orchards in calcareous soil suffer limited calcium because of lower calcium mobility in soil. The absorbed nutrient is then inadequately translocated inside the plant because the movement of calcium in the plant is very slow and mostly transmitted by xylem vessels. The addition of both Ca and B restricts transport within plants, especially under semi-arid and calcareous soil conditions (Tohidloo & Souri, 2009; Mattos et al., 2020), and these nutrients become ineffective in the soil (Manganaris et al., 2005; Mahmoudi, Akhlaghi & Forootan, 2009; Abd-Elall & Hussein, 2018). Thus, plants need a continuous supply of calcium and boron for strong vegetative growth (Del Amor & Marcelis, 2003; Fageria et al., 2009), which can be accomplished through foliar application to supplement the nutrients in the soil (Ajender, Thakur & Chawla, 2019).

Calcium, a macronutrient, is an essential element for plant growth and development and is considered an important intracellular messenger, mediating responses to hormones, stress signals, and a variety of developmental processes. Calcium also plays an important role in improving fruit set, retention, development, yield, and quality (Mahmoudi, Akhlaghi & Forootan, 2009; Hepler & Winship, 2010; Chakerolhosseini et al., 2016; Eticha et al., 2017; Wang et al., 2022), and in sustaining fruit firmness and the proliferation of vitamin C. Calcium defends membrane disorganization, protects the apparent free space of tissue generally related to senescence, and sustains the protein manufacturing capability of cells (Hussain et al., 2012; Kazemi, 2014). Calcium is extremely important in maintaining the strength of stems and stalks of plants, regulating the absorption of nutrients across plasma cell membranes, and in cell elongation and division, the structure and permeability of cell membranes, and nitrogen metabolism. Calcium stimulates the development of lignin and cellulose and the translocation and formation of carbohydrates (Aguayo, Escalona & Artés, 2008; Wang et al., 2022; Yang et al., 2023), so it is a significant factor in inflorescence and flower formation (White & Broadly, 2003). Calcium also increases the mechanical power of the cell wall because the main component of the cell wall in plants is calcium pectate, which plays a significant role in the establishment of the pedicel attachment to proximal fruit, thus reducing fruit drop (Guardiola & Garcia, 2000; El-Shafey, Abd El-Rahman & El-Azaze, 2002; Liu et al., 2019; Zaman et al., 2020).

Boron is an essential micronutrient that plays an important role in the healthy growth and development of reproductive tissues, increasing flowering, pollen grain germination, pollen tube elongation, and fruit set percentage and yield (Jehangir et al., 2017;

*Abdel-Sattar, Haikal & Hammad, 2020*). Boron also plays an active role in a variety of plant life processes, including cell division, the transfer of hormones, biosynthesis and translocation of sugars, water and nutrient uptake, tolerance of fruit crops to different disorders, the biosynthesis of IAA, and carbohydrate metabolism (*Singh, Sharma & Tyagi, 2007*; *Ahmad et al., 2009*; *Hansch & Mendel, 2009*; *Singh et al., 2016*). Boron binds strongly with cell wall constituents and helps to maintain structural integrity, though boron binds less strongly to the cell wall than calcium does (*Long & Peng, 2023*). Boron deficiency generally leads to rapid declines in tree strength, reduced vegetative and reproductive growth, death of the meristem, reduced fertility resulting in inhibition of cell expansion, limited pollen germination, and abnormal pollen tube growth, affecting fruit set and quality of fruit (*Marschner, 1995*; *Marschner, 2012*; *Bariya, Bagtharia & Patel, 2014*; *Wang et al., 2015*; *Rerkasem, Jamjod & Pusadee, 2020*).

Nitrogen compounds, carbohydrates, and hormones are involved in flower initiation, fruit set, and fruit growth. In the winter, the needed nitrogen compounds for flower initiation, blooming, and fruit set are limited due to a reduction in transpiration rate and the resulting loss in nutrients, including Ca and B, received by roots when air and soil temperatures are low (*Lovatt, Sagee & Ali, 1992*; *Malhotra, 2017*). Therefore, this study was conducted to find out whether the foliar application of seaweed extract increases leaf inflorescence formation in citrus by elevating the nutritional status of the tree, increasing growth rate, improving fruit productivity indices, and improving the quality of fruits. In addition, it is hypothesized that the application of Ca and B during the dynamic growth stages of the flower, when these nutrients may be limited, is the most effective at enhancing pollen health and ovule fertilization, as well as fruit set, fruit development, and subsequent yields (*Arrington & DeVetter, 2017*; *Long & Peng, 2023*).

This research aims to manage foliar fertilization, minimize nutrient loss, and maximize the nutrient use efficiency, enhancing the productivity and physico-chemical characteristics of Valencia oranges at harvest. Based on a literature search, this is the first extensive study into the synergistic effect of seaweed extract alone, or combined with calcium chloride ($CaCl_2$) and boric acid ($H_3BO_3$), on nutritional status, yield, and fruit quality of orange trees.

# MATERIALS AND METHODS

## Plant materials and the experimental location

The present study was conducted on 8-year-old Valencia orange trees (*Citrus sinensis*, L.) budded on sour orange rootstock (*Citrus aurantium*, L.) during two successive seasons, 2020/2021 and 2021/2022, at a private orchard in Al-Hariq town, Riyadh area, Saudi Arabia. Valencia orange trees of similar growth and vigor were planted 5 × 5 m apart under palm trees and irrigated with well water using a surface drip irrigation system. The dissolved salts in the well water were 1.87 $dSm^{-1}$ (1,197 mg/L), the adsorbed sodium was 3.46%, and the concentration of nitrate was low (0.67 mg/L). The chemical characteristics of the well water used in the study are presented in Table 1. Normal agricultural practices were applied in these orchards to all trees, according to the recommendations of the Ministry of Environmental Water & Agriculture, Saudi Arabia.

**Table 1 Physical and chemical analyses of the experimental soil and well water.**

| Parameter | Soil fractions | | | Soil class | pH | EC (dS/m) | CaCO₃ (%) | OM (%) | Soluble cations (meq/L) | | | | Soluble anions (meq/L) | | |
|---|---|---|---|---|---|---|---|---|---|---|---|---|---|---|---|
| | Sand (%) | Silt (%) | Clay (%) | | | | | | $Na^+$ | $Ca^{+2}$ | $Mg^{+2}$ | $K^+$ | $HCO^{3-}$ | $Cl^-$ | $SO4^{-2}$ |
| Soil | 75.00 | 12.00 | 13.0 | Sandy loam | 7.63 | 1.50 | 33.40 | 0.41 | 8.70 | 3.48 | 2.50 | 0.32 | 2.6 | 8.5 | 3.9 |
| Well water | —— | | | | 7.16 | 1.37 | – | – | 6.7 | 3.04 | 3.53 | 0.43 | 1.8 | 8.1 | 3.7 |

**Note:**
EC, Electrical conductivity; OM, Organic matter.

All the selected trees were similar in vigor, size, and productivity, with no visual symptoms of nutrient deficiency. The soil was sandy with a pH of 7.63. The physical and chemical characteristics of the soil at the beginning of the study are also presented in Table 1.

## Experimental design and treatments

Forty-eight trees, as uniform as possible in size, productivity, and appearance were selected for this study and were arranged in a randomized complete block design using the split-plot technique: the main plots included three levels of seaweed extract spraying (0, 5, and 10 g/L), while the subplots included four combinations of calcium chloride and/or boric acid, with four replicates × one tree/replicate per season. The trees of the main plot were sprayed with seaweed extract twice a year at flower bud differentiation (last week of November) and after 21 days from the first treatment. Each sub-main plot was then divided into subplots subjected to one of four treatments: water only, calcium chloride ($CaCl_2$) at 1 g/L, boric acid ($H_3BO_3$) at 0.5 g/L, or $CaCl_2$ at 1 g/L + $H_3BO_3$ at 0.5 g/L). These treatments were given twice a year: once at full bloom (80% flowering) and then after 21 days from the first treatment. Seaweed extract is a commercial product by Qingdao Haidelong Biotechnology Co., Ltd., Qingdao, China, containing 600–800 mg/L cytokinin and gibberellin, 16% alginic acid, 1–6% mannitol, 50% organic matter, 1% N, 16–21% $K_2O$, 0.2% Fe, 0.15% Ca, 0.2% Mg, and 1% S. Trees were sprayed using a small spraying motor (5 L solution/tree) with surfactant agent Triton B added at 0.05% to all spray solutions, including the control "tap water" treatment solution.

## Measurement of the studied parameters

Valencia orange tree productivity was evaluated in this study using the nutritional status and fruit yield parameters. Nutritional status was evaluated using the leaf mineral content and total chlorophyll content, as outlined by *Al-Dosary, Abdel-Sattar & Aboukarima (2022)*. Leaf total chlorophyll contents were estimated in three randomly sampled fresh green leaves using a Minolta (SPAD) chlorophyll meter according to the methods described by *Yadava (1986)*. The results were expressed as SPAD units. To determine leaf mineral contents, 50 mature, 7-month-old leaves were taken from non-fruiting shoots in the spring growth cycle (in mid-September) according to the methods outlined by *Summer (1985)*. Leaf samples were washed with tap water, rinsed twice in distilled water, dried in an air-drying oven at 70 °C to a constant weight, and then ground. The ground material of each sample was digested with $H_2SO_4$ and $H_2O_2$ according to the methods outlined by *Wilde et al. (1985)*. In the digested material, total nitrogen and phosphorus were

determined calorimetrically according to the methods outlined by *Evenhuis & De Waard (1980)* and *Murphy & Riley (1962)*, respectively. Potassium was determined by a flame photometer as described by *Cheng & Bray (1951)*. Calcium, magnesium, and iron were determined using a "Perkin Elmer 3300" atomic absorption spectrophotometer (*Carter, 1993*). Boron was determined spectrophotometrically at 540 nm (Model-Beckman Du 7400), according to the Azomethine-H colorimetric method described by *Wolf (1974)*.

Fruit yield parameters were fruit set, fruit retention, fruit drop, and tree yield. Two main branches from each experimental tree were selected and tagged for recording the total number of flowers at full-bloom in March and the total number of developed fruitlets after fruit set in April. These data were then used to calculate the percentage of fruit set using the following equation:

$$\text{Fruit set (\%)} = \frac{\text{Total number of fruitlets}}{\text{Total number of flowers}} \times 100 \tag{1}$$

On the same selected branches, the number of fruits retained after 1 month from the fruit set (May) was recorded and the percentages of fruit retention and fruit drop were calculated according to the following equations:

$$\text{Fruit retention \%} = \frac{\text{Total number of fruits in May}}{\text{Total number of developed fruitlets}} \times 100 \tag{2}$$

$$\text{Fruit drop (\%)} = 100 - \text{Fruit rentention} \tag{3}$$

At the ripening stage (the first week of February in both seasons), fruits from each replicate were collected and the total tree yield was determined by multiplying the average weight of fruits/tree (kg) by the number of fruits/tree (*Abdel-Sattar & Hammad, 2022*).

Samples of 20 fruits from each replicate (80 fruits for each of the applied treatments) were picked randomly at the ripening stage to determine the physical characteristics of the fruit. Fruit, pulp, and peel weight (g) were measured using an analytical balance (Mettler, Toledo, Switzerland, 0.0001 g accuracy). Fruit dimensions (length and diameter) were measured in cm using a digital caliper (Mitutoyo, Kawasaki, Japan) with a sensitivity of 0.01 mm; length/diameter ratio was also calculated (shape index). Fruit size ($cm^3$) was determined by calculating the average volume of water displaced by immersing the fruit sample in a graduated jar filled with water. Fresh fruits were ground in an electric juice extractor for freshly prepared juice, and then juice volume/fruit was evaluated (mL).

To determine the chemical characteristics of the fruit, another sample of four fruits from each replicate was randomly selected. The fruit pulp was squeezed, and the percentage of total soluble solids (TSS) in the juice was determined using a hand refractometer (Atago Co., Tokyo, Japan). Fruit acidity (%), expressed as grams of citric acid/100 mL juice, was determined by titrating the fruit juice with 0.1 N sodium hydroxide in the presence of phenolphthalein as an indicator, according to the methods of the *Association of Official Analytical Chemists (AOAC) (2019)*, and then the TSS/acidity ratio was calculated. The fruit's total sugar percentage was determined using the phenol sulfuric acid method outlined by *Malik & Singh (1980)*. The percentages of total and reducing

**Table 2 Main effects of seaweed extract sprays on the leaf mineral and chlorophyll contents of Valencia orange trees in the 2021 and 2022 seasons.**

| Seasons | Treatments | N (%) | P (%) | K (%) | Ca (%) | Mg (%) | Fe (ppm) | B (ppm) | Chlorophyll (SPAD) |
|---------|-----------|-------|-------|-------|--------|--------|----------|---------|--------------------|
| 2021 | Control | 1.22c | 0.11c | 0.69c | 2.75c | 0.24c | 63.31c | 29.56c | 38.71c |
| | 5 g/L SW | 1.70b | 0.26b | 1.11b | 3.46b | 0.39b | 90.38b | 45.81b | 55.19b |
| | 10 g/L SW | 2.25a | 0.43a | 1.41a | 4.10a | 0.54a | 120.38a | 63.56a | 65.81a |
| | LSD$_{0.05}$ | 0.029 | 0.008 | 0.031 | 0.035 | 0.008 | 1.671 | 1.151 | 0.868 |
| 2022 | Control | 1.36c | 0.12c | 0.78c | 2.83c | 0.26c | 69.19c | 30.63c | 40.02c |
| | 5 g/L SW | 1.93b | 0.28b | 1.34b | 3.59b | 0.46b | 99.25b | 51.88b | 57.76b |
| | 10 g/L SW | 2.57a | 0.51a | 1.75a | 4.28a | 0.80a | 134.56a | 81.19a | 72.61a |
| | LSD$_{0.05}$ | 0.022 | 0.010 | 0.020 | 0.027 | 0.016 | 1.969 | 1.177 | 0.193 |

**Note:**
Mean values within a column for each season that are followed by different letters are significantly different at $P \leq 0.05$.

sugars in the juice were determined according to the Lane and Eynon method, as described by *Egan, Kirk & Sawyer (1981)*. Ascorbic acid (vitamin C) was measured by the oxidation of ascorbic acid with 2, 6-dichlorophenol indophenol dye, and the results were expressed as mg/100 mL juice, as outlined by *Association of Official Analytical Chemists (AOAC) (2019)*.

## Statistical analysis

All the obtained data were subjected to two-way ANOVA using *SAS Institute Inc (2008)*. The experiment was arranged in a split-plot design in a complete randomized block system and treatment means were separated and compared using the least significant difference (LSD) at $P < 0.05$, according to the methods outlined by *Snedecor & Cochran (1990)*.

## RESULTS

### Effect of seaweed extract on productivity and physico-chemical characteristics

The whole-plot effects of spraying seaweed extract on the leaf chlorophyll and leaf mineral contents of Valencia orange trees grown in the 2021 and 2022 seasons are presented in Table 1. The nutritional status of Valencia orange trees, as measured by leaf mineral contents of N, P, K, Ca, Mg, Fe, and B, was significantly affected by SW spraying ($P < 0.05$) in both seasons (Table 2). Moreover, the chlorophyll content of the trees was significantly affected by SW spraying ($P < 0.05$) in both seasons (Table 2). The results illustrated that the foliar spray of SW at 10 g/L induced significant positive effects on leaf chlorophyll and leaf mineral contents compared with untreated control trees in both seasons.

The effects of foliar application of SW on the fruit yield parameters of the Valencia orange trees are presented in Fig. 1. Data revealed that the studied characteristics were significantly affected by the SW treatments in both seasons. Fruit set and retention percentage and yield (kg/tree) were significantly higher with 10 g/L SW treatment than with the other treatments in both seasons. The 10 g/L SW treatment also successfully reduced fruit drop. In both seasons, the 10 g/L SW treatment led to the highest values of fruit set percentage (4.51% in 2021 season and 4.98% in 2022 season), fruit retention

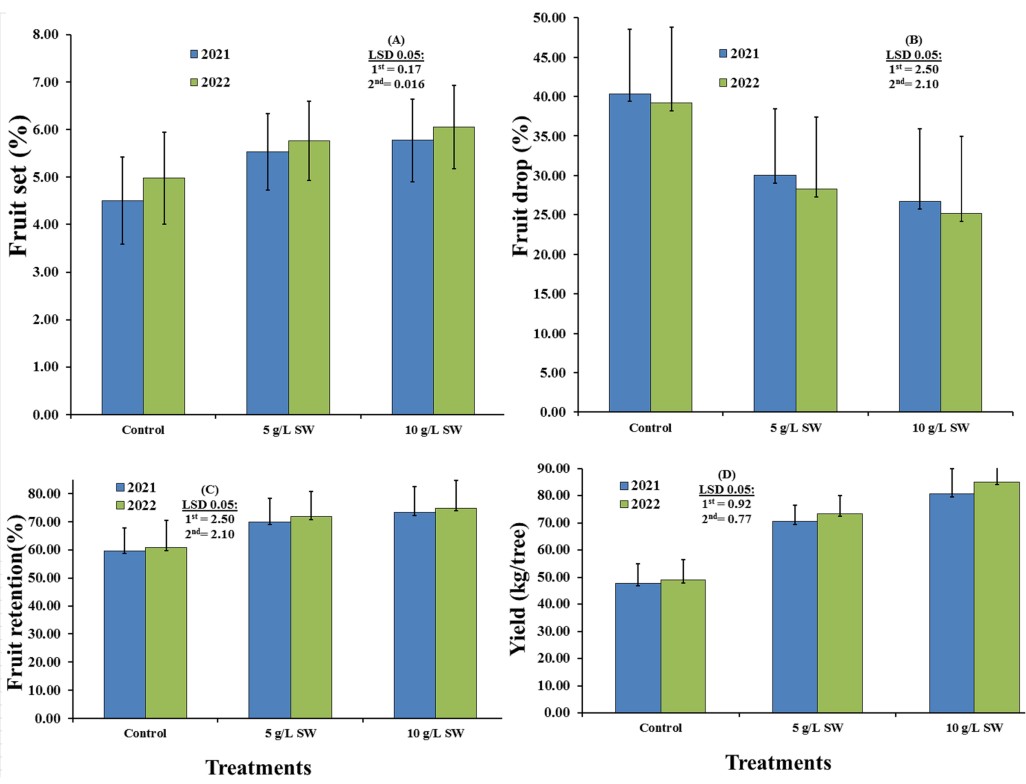

**Figure 1 Main effects of foliar seaweed extract spraying on the fruit yield parameters of Valencia orange trees in the 2021 and 2022 seasons.**

percentage (59.65% in 2021 and 60.80% in 2022), and fruit yield (47.83 kg/tree in 2021 and 48.96 kg/tree in 2022). The water-only control treatment also increased fruit drop percentage compared with the treatments, whereas increasing the SW concentration from 5 to 10 g/L increased fruit set (%), fruit retention (%), and fruit yield (kg/tree).

The whole-plot effects of seaweed extract concentration on the physical properties of Valencia orange fruit grown in the 2021 and 2022 seasons are presented in Table 3. In general, all of the measured physical properties of Valencia orange fruit were significantly affected by the SW concentration in both years ($P < 0.05$). Results from both seasons indicated that among the treatments, the 10 g/L SW application resulted in significantly higher total fruit physical properties (fruit weight, pulp weight, peel weight, fruit volume, fruit length, fruit diameter, shape index, and juice volume) and improvements in all other physical properties compared to those observed with the 5 g/L SW treatment or water-only control. The results of both seasons showed that the fruit shape index did not differ significantly with any SW application compared to the control.

The whole-plot effects of seaweed extract application on the chemical properties of Valencia orange fruit grown in the 2021 and 2022 seasons are shown in Fig. 2. All of the chemical attributes were significantly affected by the SW treatments in 2021 ($P < 0.05$), with the application of 10 g/L SW having a positive effect on all chemical properties except acidity when compared with those obtained with 5 g/L SW and in the water-only control (Fig. 2). Similar results were obtained in 2022, with 10 g/L SW significantly increasing the

**Table 3 Main effects of foliar spraying of seaweed extract on the physical characteristics of Valencia orange fruits in the 2021 and 2022 seasons.**

| Season | Treatments | Fruit weight (g) | Peel weight (g) | Pulp weight (g) | Fruit volume (cm³) | Fruit length (cm) | Fruit diameter (cm) | Shape index | Juice volume (mL) |
|---|---|---|---|---|---|---|---|---|---|
| 2021 | Control | 174.63c | 21.50c | 153.13c | 118.13c | 7.03c | 5.93c | 1.19a | 79.06c |
| | 5 g/L SW | 209.69b | 31.00b | 178.69b | 160.06b | 7.84b | 6.66b | 1.18a | 99.19b |
| | 10 g/L SW | 222.19a | 37.00a | 185.19a | 172.81a | 8.18a | 6.90a | 1.19a | 119.38a |
| | LSD$_{0.05}$ | 2.184 | 1.117 | 2.280 | 2.440 | 0.101 | 0.043 | 0.019 | 0.878 |
| 2022 | Control | 177.44c | 22.38c | 155.06c | 122.25c | 7.41c | 6.28c | 1.18a | 80.63c |
| | 5 g/L SW | 216.00b | 31.88b | 184.13b | 166.50b | 8.30b | 7.06b | 1.18a | 103.00b |
| | 10 g/L SW | 229.94a | 37.94a | 192.00a | 179.06a | 8.58a | 7.30a | 1.17a | 124.69a |
| | LSD$_{0.05}$ | 2.441 | 0.834 | 2.515 | 2.618 | 0.035 | 0.033 | 0.008 | 1.185 |

**Note:**
Mean values within a column for each season that are followed by different letters are significantly different at $P \leq 0.05$.

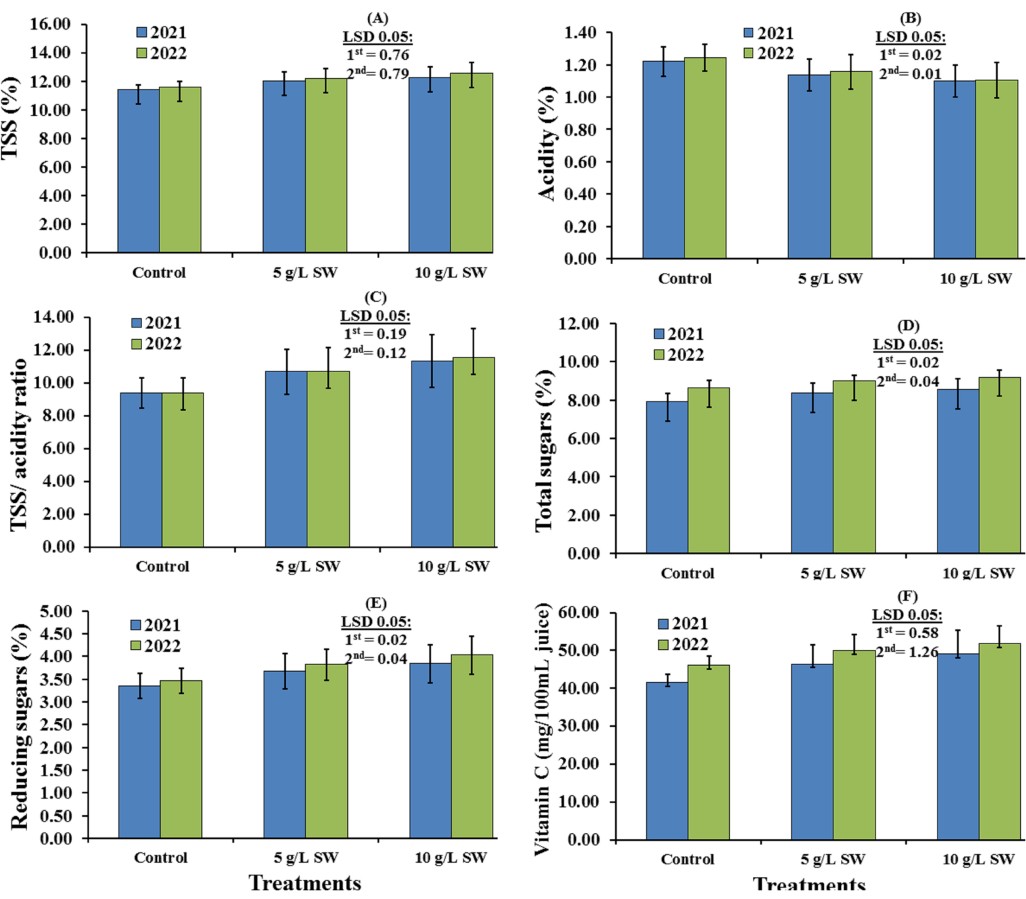

**Figure 2 (A–F) Main effects of the foliar spraying of the seaweed extract on the chemical characteristics of Valencia orange fruits in the 2021 and 2022 seasons.**

**Table 4 Sub-main effects of the foliar spraying of calcium chloride and boric acid on the nutritional status of Valencia orange trees in the 2021 and 2022 seasons.**

| Seasons | Treatments | N (%) | P (%) | K (%) | Ca (%) | Mg (%) | Fe (ppm) | B (ppm) | Chlorophyll (SPAD) |
|---|---|---|---|---|---|---|---|---|---|
| 2021 | Control | 1.52d | 0.22d | 0.93d | 3.15d | 0.33d | 79.25d | 39.17d | 47.30d |
| | 1 g/L CaCl$_2$ | 1.80b | 0.28b | 1.12b | 3.54b | 0.41b | 96.08b | 45.33c | 55.34b |
| | 0.5 g/L H$_3$BO$_3$ | 1.68c | 0.25c | 1.07c | 3.40c | 0.38c | 89.08c | 49.08b | 52.51c |
| | CaCl$_2$ + H$_3$BO$_3$ | 1.88a | 0.31a | 1.17a | 3.65a | 0.45a | 101.00a | 51.67a | 57.80a |
| | LSD$_{0.05}$ | 0.033 | 0.009 | 0.036 | 0.040 | 0.009 | 1.929 | 1.331 | 1.003 |
| 2022 | Control | 1.73d | 0.24d | 1.13d | 3.25d | 0.41d | 84.58d | 44.58d | 50.13d |
| | 1 g/L CaCl$_2$ | 2.00b | 0.32b | 1.33b | 3.63b | 0.54b | 106.58b | 52.33c | 59.36b |
| | 0.5 g/L H$_3$BO$_3$ | 1.91c | 0.28c | 1.26c | 3.53c | 0.47c | 98.08c | 57.25b | 56.05c |
| | CaCl$_2$ + H$_3$BO$_3$ | 2.16a | 0.37a | 1.43a | 3.86a | 0.59a | 114.75a | 64.08a | 61.65a |
| | LSD$_{0.05}$ | 0.025 | 0.012 | 0.022 | 0.031 | 0.018 | 2.273 | 1.360 | 0.222 |

**Note:**
Mean values within a column for each season that are followed by different letters are significantly different at $P \leq 0.05$.

values of all fruit chemical properties except for the acidity content of the fruit (Fig. 2). Across the two seasons, the 10 g/L SW application lowered fruit acidity and increased TSS (%), TSS/acidity ratio, percentages of total and reducing sugars, and vitamin C levels of Valencia orange fruit, compared to the 5 g/L SW and water-only applications.

## Effect of calcium chloride, boric acid, and their combinations on productivity and physico-chemical characteristics

The subplot effects of the foliar application of calcium chloride and boric acid on the chlorophyll and mineral contents of Valencia orange leaves in the 2021 and 2022 seasons are presented in Table 4. The data revealed that all foliar treatments significantly increased the leaf mineral and leaf chlorophyll contents compared to the control in both seasons. The calcium chloride with boric acid treatment recorded the highest values for N, P, K, Ca, Mg, Fe, and B in Valencia orange leaves in both seasons, with values of 1.88%, 0.31%, 1.17%, 3.65%, 0.45%, 101.00 and 51.6 ppm, respectively, in the first season and values of 2.16%, 0.37%, 1.43%, 3.86%, 0.59%, 114.75 and 64.08 ppm, respectively, in the second season. The 1 g/L CaCl$_2$ treatment recorded the second-highest values for all of these parameters, except for B, compared with the other foliar application treatments and control. Results from both seasons indicated that among the foliar application treatments, 1 g/L CaCl$_2$ with 0.5 g/L H$_3$BO$_3$ resulted in significantly higher leaf chlorophyll contents and improvements in all physico-chemical characteristics compared to those observed with other treatments and the control (water spray).

The subplot effects of foliar application of calcium chloride and boric acid treatments on fruit set, retention, drop percentage, and yield during the two experimental seasons are illustrated in Fig. 3. The data showed that the trees sprayed with boric acid, either alone or combined with calcium chloride, had significantly higher fruit set percentage in the first season and fruit retention percentage in both seasons when compared with the trees sprayed with other treatments. In the second season, the tree sprayed with calcium chloride and boric acid had a significantly higher fruit set percentages than those sprayed

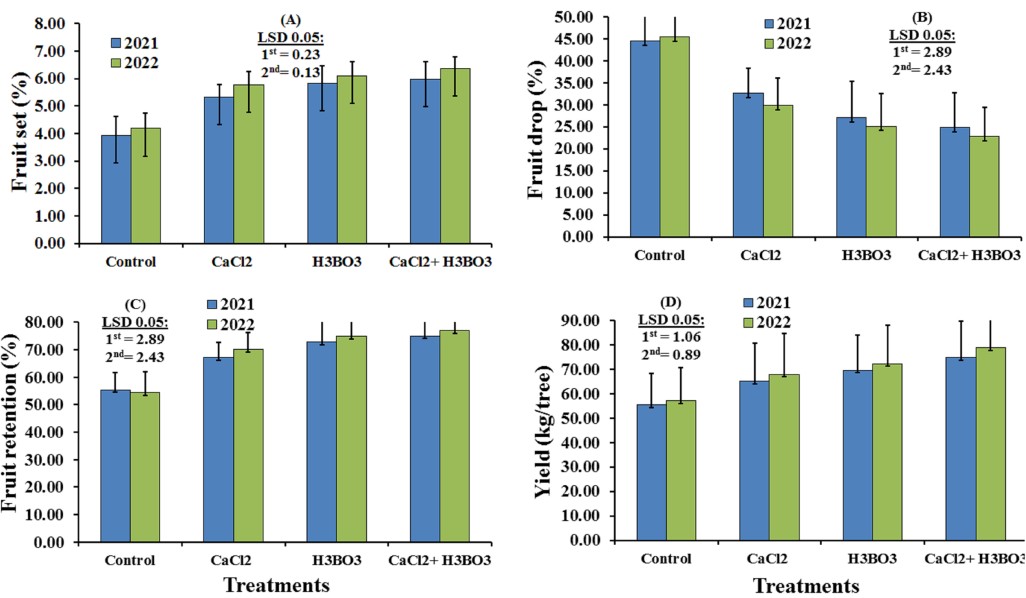

**Figure 3** (A–D) Sub-main effects of the foliar spraying of calcium chloride, boric acid, and their combinations, on the fruit yield parameters of Valencia orange in the 2021 and 2022 seasons.

with other treatments. The tree sprayed with just boric acid had the second highest fruit set percentage in the second season, while the control had the lowest fruit set percentage in both seasons. The tree sprayed with calcium chloride and boric acid also had the highest yield in both years, recorded at 74.88 and 78.91 kg/tree in the first and second seasons, respectively; these yields were significantly higher than those in all other treatments, with boric acid treatment producing the second highest yield in both seasons. In both seasons, trees sprayed with boric acid, either alone or combined with calcium chloride, had significantly lower fruit drop percentages compared to the control, with the 1 g/L $CaCl_2$ + 0.5 g/L $H_3BO_3$ treatment leading to the lowest fruit drop percentages (24.97% in 2021 and 22.90% in 2022). The water-only control recorded the highest fruit drop percentages in both seasons, at 44.55% and 45.48%, respectively.

The subplot effects of foliar application of calcium chloride and boric acid treatments on the physical properties of Valencia orange fruit in the 2021 and 2022 seasons are presented in Table 5. All of the physical characteristics except the shape index were significantly affected by the calcium chloride treatments, both alone or combined with boric acid, in the 2021 and 2022 seasons ($P < 0.05$). The experimental treatments significantly increased the average fruit weight, peel weight, pulp weight, fruit volume, fruit length, fruit diameter, and juice volume relative to those measures in the control fruits. Additionally, in both seasons, 1 g/L $CaCl_2$ + 0.5 g/L $H_3BO_3$ produced the highest values of all treatments, followed by the 1 g/L $CaCl_2$ treatment, for juice volume, fruit weight, pulp weight, peel weight, fruit length, fruit volume, and fruit diameter. All treatments slightly increased the shape index (fruit length/fruit diameter), however, the differences among all treatments and the control were not large enough to be significant.

**Table 5 Sub-main effects of the foliar spraying of calcium chloride and boric acid on the physical characteristics of Valencia orange fruits in the 2021 and 2022 seasons.**

| Seasons | Treatments | Fruit weight (g) | Peel weight (g) | Pulp weight (g) | Fruit volume (cm$^3$) | Fruit length (cm) | Fruit diameter (cm) | Shape index | Juice volume (mL) |
|---|---|---|---|---|---|---|---|---|---|
| 2021 | Control | 184.33d | 25.33d | 159.00c | 132.42d | 7.29c | 6.15d | 1.19a | 92.17d |
| | 1 g/L CaCl$_2$ | 208.50b | 31.67b | 176.83ab | 156.83b | 7.76b | 6.60b | 1.18a | 101.42b |
| | 0.5 g/L H$_3$BO$_3$ | 203.67c | 28.67c | 175.00b | 150.00c | 7.74b | 6.49c | 1.19a | 97.17c |
| | CaCl$_2$ + H$_3$BO$_3$ | 212.17a | 33.67a | 178.50a | 162.08a | 7.95a | 6.75a | 1.18a | 106.08a |
| | LSD$_{0.05}$ | 2.521 | 1.289 | 2.632 | 2.817 | 0.116 | 0.05 | 0.023 | 1.013 |
| 2022 | Control | 188.33d | 26.25d | 162.08c | 137.42d | 7.68d | 6.54d | 1.17a | 95.00d |
| | 1 g/L CaCl$_2$ | 212.83b | 32.50b | 180.33b | 162.50b | 8.23b | 6.97b | 1.18a | 105.33b |
| | 0.5 g/L H$_3$BO$_3$ | 208.17c | 29.67c | 178.50b | 155.92c | 8.13c | 6.90c | 1.18a | 100.33c |
| | CaCl$_2$ + H$_3$BO$_3$ | 221.83a | 34.50a | 187.33a | 167.92a | 8.33a | 7.10a | 1.17a | 110.42a |
| | LSD$_{0.05}$ | 2.819 | 0.963 | 2.904 | 3.023 | 0.041 | 0.039 | 0.009 | 1.368 |

**Note:**
Values within a column with the same letter(s) are not significantly different, according to LSD ($P < 0.05$).

The subplot effects of foliar application of calcium chloride and boric acid on the chemical properties of Valencia orange fruit in the 2021 and 2022 seasons are presented in Fig. 4. In general, all of the chemical properties were significantly affected by boric acid treatment, either alone or combined with calcium chloride, in both growing seasons ($P < 0.05$), with all chemical characteristics increasing significantly except acidity, which decreased significantly. The data indicated that in both seasons, the 1 g/L CaCl$_2$ + 0.5 g/L H$_3$BO$_3$ treatment produced the highest fruit TSS content, TSS/acidity, vitamin C, reducing sugar content, and total sugar contents, with values of 12.47%, 11.74, 50.17 mg/100 mL$^{-1}$ juice, 3.96%, and 8.74%, respectively, in the first season, and 12.73%, 11.92, 52.33 mg/100 mL$^{-1}$ juice, 4.10%, and 9.27%, respectively, in the second season. The 1 g/L CaCl$_2$ + 0.5 g/L H$_3$BO$_3$ treatment also produced the lowest acidity value in the first (1.07%) and second (1.08%) seasons.

## Interaction effects of seaweed extract application with calcium chloride and boric acid on productivity and physico-chemical characteristics of Valencia orange trees

The interaction effects between SW combined with calcium chloride and boric acid on the nutritional status, in terms of leaf mineral constituents (nitrogen, phosphorus, potassium, calcium, and magnesium), and on the total leaf chlorophyll content of Valencia orange trees grown in the 2021 and 2022 seasons are shown in Table 6. The mineral content of nitrogen, phosphorus, potassium, calcium, and magnesium, as well as leaf iron and boron, and total chlorophyll content (SPAD) differed significantly among the treatments (as indicated in Table 6). In both seasons, the trees sprayed with 10 g/L SW combined with 1 g/L calcium chloride and 0.5 g/L boric acid had the highest N, P, K, Ca, Mg, Fe, B, and chlorophyll content (SPAD), with values of 2.45%, 0.48%, 1.50%, 4.33%, 0.61%, 128.00,

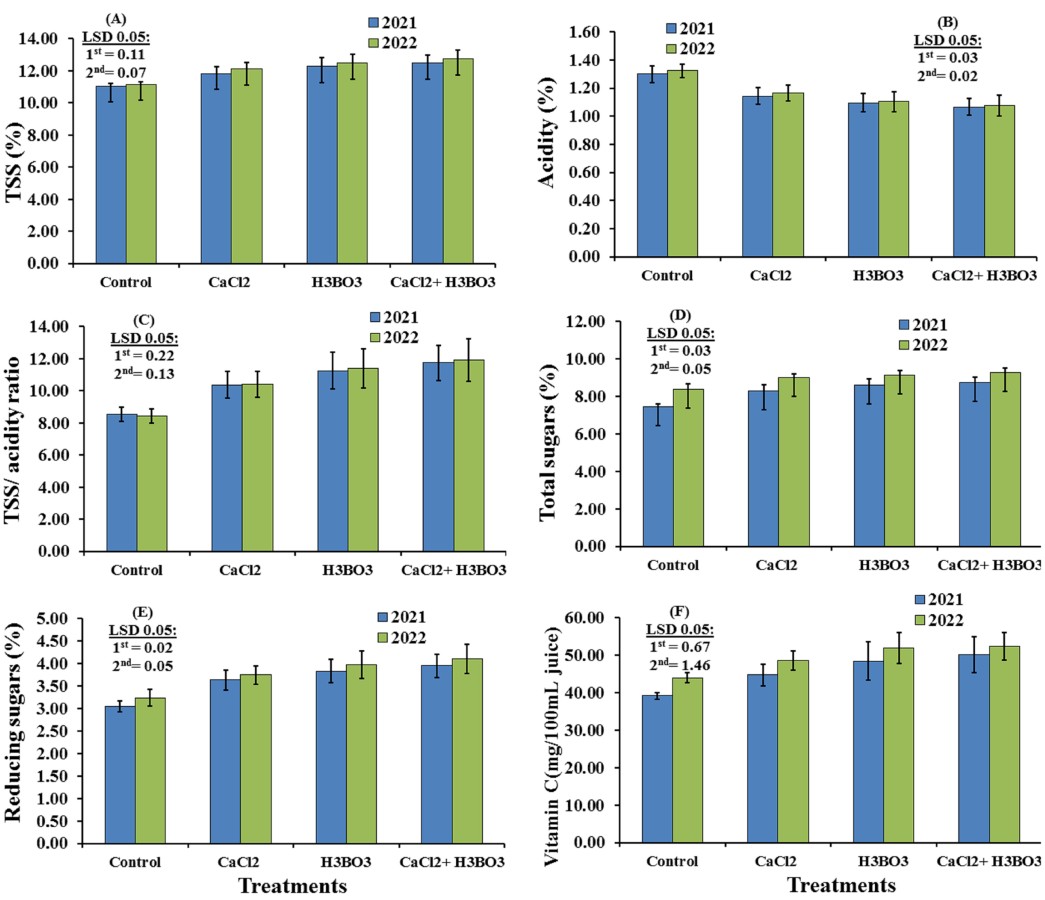

**Figure 4 (A–E)** Sub-main effects of the foliar spraying of calcium chloride, boric acid, and their combinations on the chemical characteristics of Valencia orange fruits in the 2021 and 2022 seasons.

70.00 ppm, and 69.33 SPAD, respectively, in the first season and 2.79%, 0.57%, 1.85%, 4.55%, 0.90%, 145.50, 96.75 ppm, and 75.98 SPAD, respectively, in the second season (Table 6). Moreover, the B content was highest in the tree sprayed with 10 g/L SW combined with 0.5 g/L boric acid, with values of 66.00 in 2021 and 88.50 in 2022 (Table 6), followed by the tree sprayed with 10 g/L SW + CaCl$_2$ with H$_3$BO$_3$. In both seasons, the control treatment produced significantly lower mineral contents and total leaf chlorophyll content than the other treatments.

The interaction effects of seaweed extract application with calcium chloride and boric acid on the fruit set (%), fruit drop (%) fruit retention (%), and fruit yield (kg/tree) of Valencia orange trees in the 2021 and 2022 seasons are presented in Fig. 5. In both seasons, all treatments significantly improved the fruit set percentage, fruit retention percentage, and yield (kg/tree) compared with the water-only control. However, the most effective treatment was 10 g/L SW combined with 1 g/L calcium chloride and 0.5 g/L boric acid, which induced the highest fruit set percentage, fruit retention percentage, and yield (kg/tree), followed by 10 g/L SW combined with 0.5 g/L boric acid (Figs. 5A, 5B, 5E–5H). The percentages of fruit set, fruit retention, and yield reached maximum values of 6.5%,

**Table 6 Interaction effects of foliar seaweed extract treatment combined with calcium chloride and boric acid on the nutritional status of Valencia orange trees in the 2021 and 2022 seasons.**

| Seasons | Treatments | N (%) | P (%) | K (%) | Ca (%) | Mg (%) | Fe (ppm) | B (ppm) | Chlorophyll (SPAD) |
|---|---|---|---|---|---|---|---|---|---|
| 2021 | Control | 1.02 | 0.08 | 0.50 | 2.30 | 0.16 | 45.50 | 21.00 | 29.40 |
| | 1 g/L $CaCl_2$ | 1.30 | 0.12 | 0.75 | 2.91 | 0.28 | 70.00 | 29.00 | 41.60 |
| | 0.5 g/L $H_3BO_3$ | 1.20 | 0.11 | 0.70 | 2.80 | 0.24 | 63.00 | 33.25 | 39.13 |
| | $CaCl_2$ + $H_3BO_3$ | 1.36 | 0.14 | 0.80 | 3.00 | 0.30 | 74.75 | 35.00 | 44.73 |
| | 5 g/L SW | 1.54 | 0.21 | 1.00 | 3.31 | 0.35 | 80.25 | 40.25 | 50.60 |
| | 5 g/L SW + $CaCl_2$ | 1.75 | 0.28 | 1.15 | 3.51 | 0.40 | 95.50 | 45.00 | 56.98 |
| | 5 g/L SW + $H_3BO_3$ | 1.65 | 0.24 | 1.10 | 3.40 | 0.38 | 85.50 | 48.00 | 53.73 |
| | 5 g/L SW + $CaCl_2$ + $H_3BO_3$ | 1.85 | 0.32 | 1.20 | 3.61 | 0.43 | 100.25 | 50.00 | 59.45 |
| | 10 g/L SW | 2.00 | 0.37 | 1.30 | 3.85 | 0.47 | 112.00 | 56.25 | 61.90 |
| | 10 g/L SW + $CaCl_2$ | 2.35 | 0.46 | 1.47 | 4.22 | 0.56 | 122.75 | 62.00 | 67.45 |
| | 10 g/L SW $_3$ + $H_3BO_3$ | 2.20 | 0.42 | 1.40 | 4.00 | 0.52 | 118.75 | 66.00 | 64.67 |
| | 10 g/L SW + $CaCl_2$ + $H_3BO_3$ | 2.45 | 0.48 | 1.50 | 4.33 | 0.61 | 128.00 | 70.00 | 69.23 |
| | $LSD_{0.005}$ | 0.057 | 0.015 | 0.062 | 0.070 | 0.016 | 3.341 | 2.305 | 1.737 |
| 2022 | Control | 1.20 | 0.09 | 0.57 | 2.35 | 0.18 | 50.25 | 24.25 | 30.03 |
| | 1 g/L $CaCl_2$ | 1.39 | 0.14 | 0.86 | 2.96 | 0.29 | 76.00 | 30.00 | 44.00 |
| | 0.5 g/L $H_3BO_3$ | 1.30 | 0.11 | 0.75 | 2.85 | 0.24 | 64.75 | 33.25 | 40.10 |
| | $CaCl_2$ + $H_3BO_3$ | 1.54 | 0.16 | 0.95 | 3.17 | 0.32 | 85.75 | 35.00 | 45.95 |
| | 5 g/L SW | 1.70 | 0.22 | 1.18 | 3.40 | 0.37 | 80.25 | 42.50 | 51.98 |
| | 5 g/L SW + $CaCl_2$ | 1.96 | 0.29 | 1.37 | 3.61 | 0.47 | 107.00 | 54.50 | 60.05 |
| | 5 g/L SW + $H_3BO_3$ | 1.88 | 0.24 | 1.30 | 3.52 | 0.43 | 96.75 | 50.00 | 55.98 |
| | 5 g/L SW + $CaCl_2$ + $H_3BO_3$ | 2.16 | 0.37 | 1.50 | 3.85 | 0.57 | 113.00 | 60.50 | 63.03 |
| | 10 g/L SW | 2.30 | 0.43 | 1.65 | 4.00 | 0.68 | 123.25 | 67.00 | 68.38 |
| | 10 g/L SW + $CaCl_2$ | 2.63 | 0.53 | 1.77 | 4.33 | 0.86 | 136.75 | 72.50 | 74.03 |
| | 10 g/L $SW_3$ + $H_3BO_3$ | 2.55 | 0.49 | 1.72 | 4.22 | 0.75 | 132.75 | 88.50 | 72.08 |
| | 10 g/L SW + $CaCl_2$ + $H_3BO_3$ | 2.79 | 0.57 | 1.85 | 4.55 | 0.90 | 145.50 | 96.75 | 75.98 |
| | $LSD_{0.005}$ | 0.044 | 0.020 | 0.040 | 0.054 | 0.031 | 3.938 | 2.354 | 0.385 |

82.05%, and 90.94 kg/tree, respectively, in 2021 and 6.81%, 83.30%, and 97.32 kg/tree, respectively, in 2022. In both seasons, the water-only control had a significantly lower fruit set percentage, fruit retention percentage, and yield than those seen with other treatments. Data from both seasons showed that all treatments significantly decreased the percentage of fruit drop compared to the control. The control had a significantly higher fruit drop percentage followed by 5 g/L SW + water spray, while the 10 g/L SW + $CaCl_2$ with $H_3BO_3$ treatment had the lowest fruit drop percentage in both seasons (Figs. 5C and 5D).

The two-way interactions between seaweed extract combined with calcium chloride and boric acid sprays on the physical properties of Valencia orange fruit grown in the 2021 and 2022 seasons are presented in Table 7. As shown in Table 7, all physical properties of the fruit that were measured in this study were significantly affected by SW combined with calcium chloride and boric acid sprays. Analysis of variance revealed significant differences among the treatments for fruit weight (g), peel weight (g), pulp weight (g), fruit volume

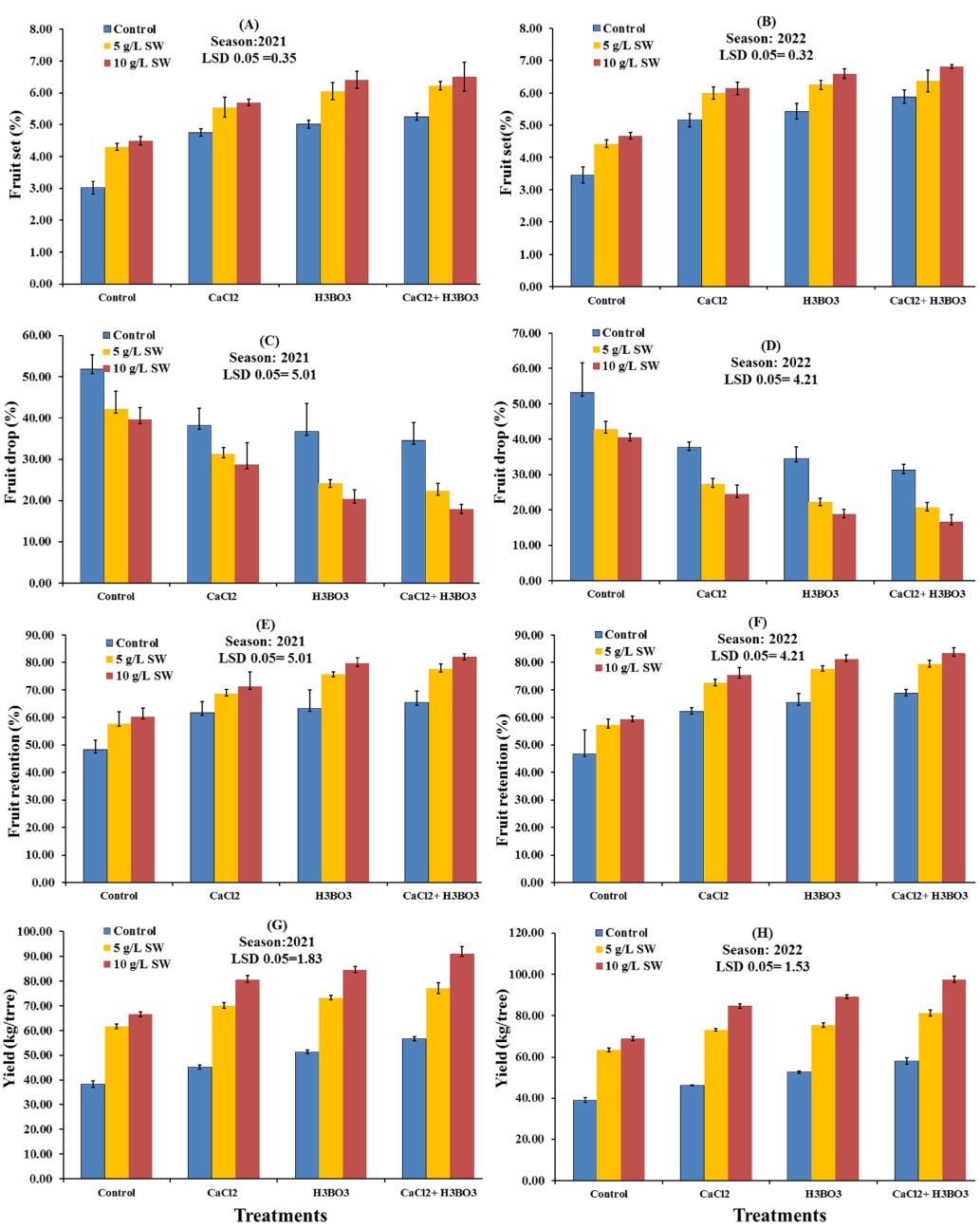

**Figure 5 (A–H) Interaction influences of foliar seaweed extract spraying combined with calcium chloride and boric acid on the fruit yield parameters of Valencia orange during the 2021 and 2022 seasons.**

(cm³), fruit length (cm), fruit diameter (cm), shape index, and juice volume (cm³; Table 7). In both seasons, fruit from all treated trees had significantly higher physical properties than those from the water-only control (Table 7). The 10 g/L SW treatment combined with 0.5 g/L boric acid and 1 g/L calcium chloride treatment resulted in the highest fruit weight, pulp weight, peel weight, fruit volume, fruit length, fruit diameter, shape index, and juice volume, followed by the 10 g/L SW + 1 g/L calcium chloride treatment. There were no

**Table 7 Interaction effects of foliar seaweed extract spraying combined with calcium chloride and boric acid on the physical characteristics of Valencia orange fruits in the 2021 and 2022 seasons.**

| Seasons | Treatments | Fruit weight (g) | Peel weight (g) | Pulp weight (g) | Fruit volume (cm³) | Fruit length (cm) | Fruit diameter (cm) | Shape index | Juice volume (mL) |
|---|---|---|---|---|---|---|---|---|---|
| 2021 | Control | 151.75 | 18.00 | 133.75 | 102.00 | 6.43 | 5.40 | 1.19 | 71.25 |
| | 1 g/L CaCl₂ | 183.25 | 23.00 | 160.25 | 125.50 | 7.24 | 6.06 | 1.20 | 81.75 |
| | 0.5 g/L H₃BO₃ | 177.00 | 20.00 | 157.00 | 115.00 | 7.15 | 5.98 | 1.20 | 77.75 |
| | CaCl₂ +H₃BO₃ | 186.50 | 25.00 | 161.50 | 130.00 | 7.30 | 6.28 | 1.16 | 85.50 |
| | 5 g/L SW | 198.75 | 28.00 | 170.75 | 145.00 | 7.60 | 6.44 | 1.18 | 92.25 |
| | 5 g/L SW + CaCl₂ | 214.50 | 32.00 | 182.50 | 165.00 | 7.75 | 6.69 | 1.16 | 101.25 |
| | 5 g/L SW + H₃BO₃ | 210.00 | 30.00 | 180.00 | 160.00 | 7.88 | 6.70 | 1.18 | 97.00 |
| | 5 g/L SW + CaCl₂ + H₃BO₃ | 215.50 | 34.00 | 181.50 | 170.25 | 8.15 | 6.81 | 1.20 | 106.25 |
| | 10 g/L SW | 202.50 | 30.00 | 172.50 | 150.25 | 7.83 | 6.60 | 1.19 | 113.00 |
| | 10 g/L SW + CaCl₂ | 227.75 | 40.00 | 187.75 | 180.00 | 8.30 | 7.05 | 1.18 | 121.25 |
| | 10 g/L SW₃ + H₃BO₃ | 224.00 | 36.00 | 188.00 | 175.00 | 8.20 | 6.79 | 1.21 | 116.75 |
| | 10 g/L SW + CaCl₂ + H₃BO₃ | 234.50 | 42.00 | 192.50 | 186.00 | 8.41 | 7.15 | 1.18 | 126.50 |
| | LSD₀.₀₀₅ | 4.367 | 2.233 | 4.559 | 4.880 | 0.201 | 0.087 | 0.039 | 1.755 |
| 2022 | Control | 155.00 | 18.75 | 136.25 | 104.50 | 6.83 | 5.81 | 1.18 | 72.00 |
| | 1 g/L CaCl₂ | 185.75 | 23.75 | 162.00 | 129.75 | 7.60 | 6.40 | 1.19 | 83.75 |
| | 0.5 g/L H₃BO₃ | 180.00 | 21.00 | 159.00 | 119.75 | 7.50 | 6.30 | 1.19 | 79.25 |
| | CaCl₂ +H₃BO₃ | 189.00 | 26.00 | 163.00 | 135.00 | 7.70 | 6.60 | 1.17 | 87.50 |
| | 5 g/L SW | 203.00 | 29.00 | 174.00 | 151.25 | 8.00 | 6.80 | 1.18 | 96.50 |
| | 5 g/L SW + CaCl₂ | 220.00 | 32.75 | 187.25 | 171.50 | 8.40 | 7.12 | 1.18 | 105.25 |
| | 5 g/L SW + H₃BO₃ | 215.50 | 31.00 | 184.50 | 166.75 | 8.30 | 7.10 | 1.17 | 100.00 |
| | 5 g/L SW + CaCl₂ +H₃BO₃ | 225.50 | 34.75 | 190.75 | 176.50 | 8.50 | 7.21 | 1.18 | 110.25 |
| | 10 g/L SW | 207.00 | 31.00 | 176.00 | 156.50 | 8.20 | 7.00 | 1.17 | 116.50 |
| | 10 g/L SW + CaCl₂ | 232.75 | 41.00 | 191.75 | 186.25 | 8.70 | 7.40 | 1.18 | 127.00 |
| | 10 g/L SW₃ + H₃BO₃ | 229.00 | 37.00 | 192.00 | 181.25 | 8.60 | 7.30 | 1.18 | 121.75 |
| | 10 g/L SW + CaCl₂ + H₃BO₃ | 251.00 | 42.75 | 208.25 | 192.25 | 8.80 | 7.50 | 1.17 | 133.50 |
| | LSD₀.₀₀₅ | 4.882 | 1.669 | 5.029 | 5.235 | 0.071 | 0.067 | 0.016 | 2.370 |

significant differences in fruit shape index between all treatments in both the first season and second season.

The interaction effects of seaweed extract combined with calcium chloride and boric acid sprays on the chemical characteristics of Valencia orange juice in the 2021 and 2022 seasons are shown in Figs. 6 and 7. In both seasons, the juice from all treated trees had significantly higher chemical properties, except acidity content, than the juice from the untreated control. In 2021, the highest values for all chemical properties, except acidity, were observed with the 10 g/L SW combined with 0.5 g/L boric acid and 1 g/L calcium

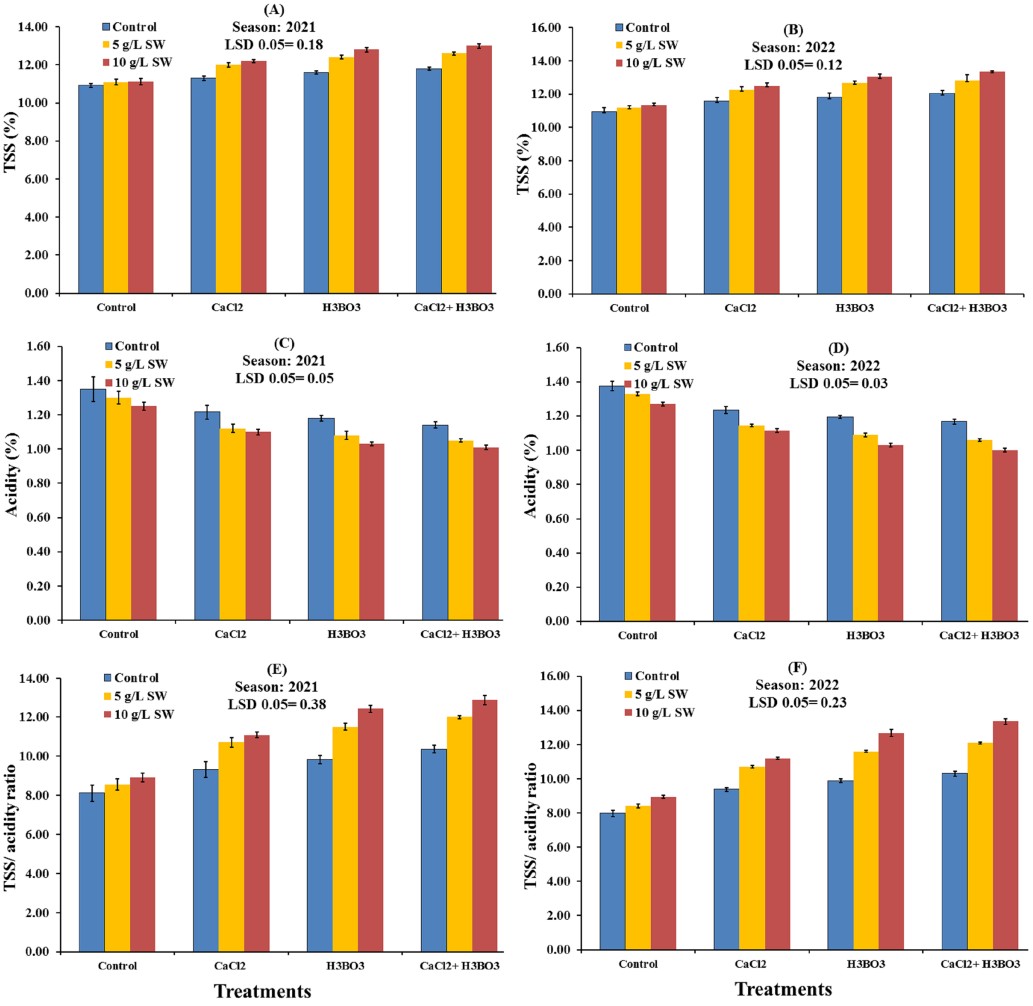

**Figure 6 (A–F) Interaction influences of foliar seaweed extract spraying combined with calcium chloride and boric acid on the TSS, acidity, and TSS/acidity ratio in the juice of Valencia orange fruits during the 2021 and 2022 seasons.**

chloride treatment, followed by the 10 g/L SW treatment, and then the 5 g/L SW + 1 g/L calcium chloride + 0.5 g/L boric acid treatment. In 2021, the highest values of TSS content, TSS/acidity, vitamin C, reducing sugar content, and total sugar content in the juice of Valencia oranges were recorded as 12.80%, 12.43, 53.75 mg/100 mL$^{-1}$ juice, 4.12%, and 8.93%, respectively. In the second season (2022), the highest values for all chemical properties, except acidity, were also observed with the 10g/L SW combined with 0.5 g/L boric acid and 1 g/L calcium chloride treatment, followed by the g/L SW + 0.5 g/L boric acid treatment (Figs. 6 and 7). The highest values of vitamin C, TSS content, TSS/acidity, and total and reducing sugar contents in the juice of Valencia oranges were 56.00 mg/100 mL$^{-1}$ juice, 13.35%, 13.35, 9.55%, and 4.45%, respectively, in the second season. The fruit from the water-only control had significantly higher fruit acidity than the fruits from all treated trees. In addition, the 10 g/L SW with 0.5 g/L boric acid and 1 g/L calcium chloride

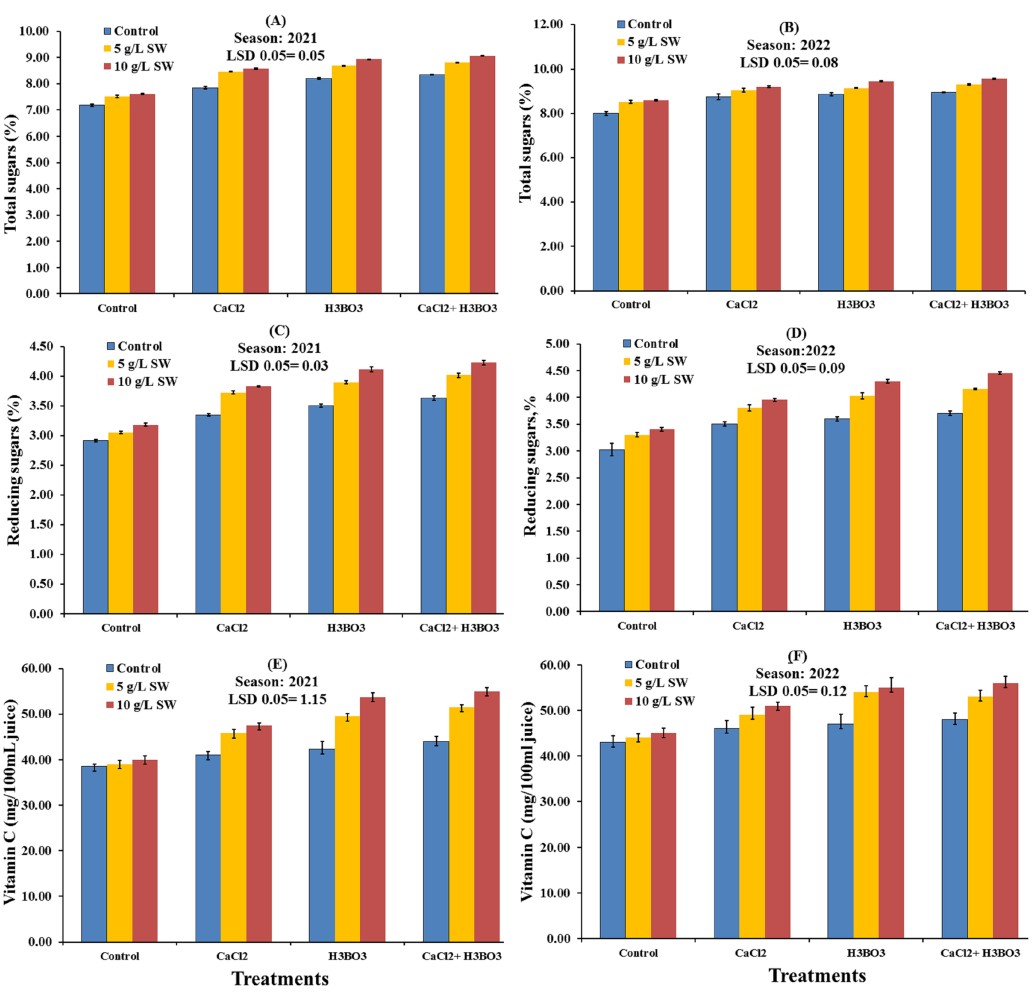

**Figure 7** (A–F) Interaction influences of foliar seaweed extract spraying combined with calcium chloride and boric acid on total sugar, reducing sugar, and vitamin C contents in the juice of Valencia orange fruits during the 2021 and 2022 seasons.

treatment significantly decreased the total acidity (TA) compared to the other treatments in both seasons.

## DISCUSSION

The physiological effect of seaweed extract application is primarily responsible for enhancing the effect of boric acid and calcium chloride on the physico-chemical properties of Valencia orange trees, which affected the quality of the fruit. In addition, the increase in total leaf chlorophyll content and leaf mineral contents resulted in an increase in photosynthetic rate and carbohydrate reserves, which improved fruit yield parameters and fruit quality. In general, the foliar spray of SW with calcium chloride, boric acid, and their combinations, increased leaf chlorophyll, leaf mineral contents, fruit set, fruit retention, fruit drop, fruit yield, fruit weight (g), peel weight (g), pulp weight (g), fruit volume (cm3), fruit length (cm), fruit diameter (cm), juice volume (cm3), TSS content, TSS/acidity ratio, vitamin C, reducing sugar content, and total sugar content, and decreased fruit drop and

fruit acidity, with most properties changing significantly compared with the control. These results are in line with those of *Galeriani et al. (2022)*, which indicate that the foliar application of calcium (Ca) and boron (B) is widely used because Ca and B are essential nutrients for the growth and development of plants, in addition to regulating various physiological processes. *Leite et al. (2011)* and *Bezerra et al. (2023)* found that calcium and boron are important nutrients in the stages of flowering, pollen germination, pollen tube growth, fruit set, and flower fertilization. Additionally, several studies have shown that the combined application of Ca + B promotes greater vegetative development, increased yield, fruit quality, and productivity in different crops compared to the isolated application of these elements (*Singh, Sharma & Tyagi, 2007*; *Bhatt et al., 2012*; *Meena et al., 2016*; *Hikal, Ibrahim & Abdelaziz, 2017*; *Meena et al., 2017*; *Al-Mayahi, 2020*).

## Nutritional status

This study found that spraying seaweed extract improved the leaf chlorophyll contents and leaf composition of nitrogen, phosphorus, potassium, calcium, magnesium, iron, and boron compared to the untreated control. The enhanced effect of seaweed on leaf chemical constituents and leaf chlorophyll contents of Valencia orange might be attributed to seaweed's own higher content of mineral elements like N, P, K, Mg, Ca, S, Cu, Fe, Mn, B, and Mo, and of amino acids, vitamins and antioxidants and natural plant hormones, such as IAA, GA3, and cytokinins (*Tung-Yunn et al., 2003*; *Harhash et al., 2021*), which stimulate plant growth, improve chlorophyll and photosynthesis processes, and have many other beneficial effects on plant growth and development (*Abd El-Moniem & Abd-Allah, 2008*; *Mosa et al., 2023*). Seaweed plays a crucial role in improving the photosynthesis rate and enhancing plant uptake of mineral elements such as phosphorous and nitrogen that are directly linked to chlorophyll formation in tree leaves (*Zhang & Ervin, 2004*; *Al-Saif et al., 2023*; *Mosa et al., 2023*; *Yang et al., 2023*). Additionally, SW is a natural and cheap way to improve the resistance of plants to unfavorable stresses, improve plant growth hormones, increase plant cell division (*Fornes, Sanchez & Guardiola, 2002*; *Spinelli et al., 2009*; *Prasad et al., 2010*; *Battacharyya et al., 2015*; *Harhash et al., 2021*), and improve shoot length and leaf area (*Colavita et al., 2011*). Seaweed also contains magnesium, which is a crucial element for chlorophyll synthesis (*Al-Saif et al., 2023*). Similar to the results of this study, *Salama (2015)* and *El-Badawy (2017)* reported that spraying SW had a highly positive effect on the nutritional status of Valencia orange trees. The foliar application of SW at low concentrations offers a quick method of supplying nutrients to plants, stimulating a variety of physiological plant responses, and improving nutritional status (*Rana et al., 2023*).

A symbiotic interaction exists between calcium and boron, with the effect of one dependent on the other. In addition, plants need boron to use calcium efficiently. This means that calcium treatments are not nearly as successful if the crop does not receive enough boron in the tissue (*Marschner, 2002*). Calcium and boron addition improves the uptake of nutrients, improving the nutritional status of Valencia orange tree leaves in terms of mineral elements and total chlorophyll contents. Calcium accumulation in the leaves increases mineral content, which may help enhance cell division and root growth,

improving nutrient absorption (*Rosen et al., 2006*). Calcium regulates the absorption of nutrients across plasma cell membranes and is an important factor in flower formation because of its role in plant cell elongation and division, the structure and permeability of cell membranes, nitrogen metabolism, and carbohydrate translocation (*White & Broadly, 2003*). *Polevoiy (1989)* reported that calcium promotes early root formation and growth, improves general plant vigor, and enhances the uptake of other nutrients such as phosphorous, manganese, iron, zinc, and boron. These nutrients are involved in chlorophyll biosynthesis, so Ca application leads to increased leaf chlorophyll content.

The beneficial effects of boron on the growth and nutritional status (as measured by leaf mineral contents and leaf chlorophyll content) of trees might be attributed to boron's ability to stimulate the biosynthesis, building, and translocation of sugars, and boron's ability to promote cell division, root development, the building of natural hormones, and nutrient and water uptake (*Mengel & Kirkby, 2001*). Similar results were obtained by *Ibrahim & Al-Wasfy (2014)*, who reported that exogenously applying 0.05% boric acid on Valencia orange trees significantly improved the leaf area and the leaf contents of chlorophyll a, chlorophyll b, total carotenoids, total chlorophylls, nitrogen, phosphorous, potassium, and magnesium compared to the control treatment. Similarly, *Harhash, Nasr Alla & Mosa (2022)* showed that the foliar application of boron on Kallamata olive trees was extremely effective in enhancing the nutritional status of the trees, greatly improving the leaf chlorophyll content and the leaf content of potassium, phosphorous, nitrogen, zinc and boron.

## Yield parameters

Spraying Valencia orange trees with SW combined with boric acid and/or calcium chloride had a significant effect on the fruit yield parameters of the trees when compared with the untreated oranges, improving the flower count, fruit set percentages, fruit numbers, and fruit yield, and reducing fruit drop. This may be because the algae extract contains minerals, vitamins, and growth regulators, especially IAA and cytokinins, which improve plant productivity (*Stirck et al., 2003*; *Battacharyya et al., 2015*; *Ali, Ramsubhag & Jayaraman, 2021*). Seaweed's positive effect on fruit yield parameters might also be attributed to improved chlorophyll production and photosynthesis processes and their essential role in balancing the ratio between carbohydrates and nitrogen to support flowering, leading to an increase in the number of fruits per tree (*Neumann & ZurNieden, 2001*; *Salama, 2015*). The foliar spraying of SE encourages leaf and shoot growth, extends blooming time, and improves the quality of flower formation, which all improve fruit set percentage and productivity (*Basak, 2008*). These results are consistent with those obtained by *Fornes, Sanchez & Guardiola (2002)* on "de Nules" Clementine mandarin and "Navelina" orange; *Ahmed et al. (2013)* and *Salama (2015)* on Valencia orange, and *Mosa et al. (2023)* on apple cv. Anna. Treating oranges with SW increased the fruit yield parameters and reduced the fruit drop percentage (*Arioli, Mattner & Winberg, 2015*).

The results of this study clearly showed that spraying trees with calcium chloride and boric acid, either alone or combined, had a positive effect on yield parameters such as fruit set, fruit drop, fruit retention, overall tree yield, and fruit retention of Valencia orange trees

compared to control. The increase in yield of Valencia orange fruits seen with the application of calcium and boron treatments may be due to the synergistic effect of boron and calcium nutrients, as they are both directly and indirectly involved in many physiological processes and enzyme activities (*Meena et al., 2016*; *Sheikh et al., 2021*). The reduced fruit drop and increased fruit retention and fruit number recorded as a result of the effect of these treatments may be due to boron and calcium obstructing enzymes such as polygalacturonase from reaching their active sites, thereby retarding the occurrence of abscission zones (*Marcelis et al., 2004*). The reduction of fruit abscission may also be due to the initiation of enzymatic antioxidants in fruit petiole (*Ali, Elhamahmy & El-Shiekh, 2017*). There is a correlation between fruit drop and the internal hormonal levels in the plant system: as the level of internal auxin concentration in the plant system increases, the fruit retention capacity increases, leading to an increase in the number of fruits per plant. Boron and calcium are required for pollen grain germination and pollen tube elongation (*Krichevsky et al., 2007*), which aids in effective fertilization or pollination and prevents blossom abortion. The effects of Ca and B application, however, are dependent on a balance of B and Ca levels in the plant (*Zoz et al., 2016*). Foliar application of calcium and boron nutrients can enhance photosynthesis with structural and reproductive functions and increase the setting of flowers, thereby increasing productivity (*Galeriani et al., 2022*).

Boron, as an essential micronutrient, plays an important role in increasing pollen grain germination and pollen tube elongation (*Ganie et al., 2013*) through the formation of the boron-sorbitol complex, which enhances absorption, translocation, and metabolism of sugar in pollen and by synthesizing the pectin material for the cell wall of the growing pollen tube (*Bastakoti et al., 2022*), increasing fruit set percentage, fruit retention percentage, and yield. Boron's active involvement in the biosynthesis of auxins may also have controlled the fruit drop and increased fruit set and retention (*Singh & Maurya, 2003*). Boron is also associated with hormonal metabolism, photosynthate accumulation, and water relations, thereby increasing fruit retention (*Sankar, Saraladevi & Parthiban, 2013*). *Bariya, Bagtharia & Patel (2014)* found that reproductive growth, mainly flowering, fruit set, and yield, is particularly sensitive to boron deficiency, since boron is very important for all reproductive tissues. During flowering and fruit setting, boron deficiency can result in the dropping of flowers and poor fruit setting (*Marschner, 2012*). Although B is readily absorbed and mobile within the xylem of plants, foliar application of B is preferred over soil application because of the relatively narrow range from deficient to toxic levels of boron (*Trautmann et al., 2014*). In addition, foliar applications during the dynamic growth phase, occurring during floral development, pollen germination, fertilization, and early fruit development, may increase absorption and movement to targeted tissues (*Brown & Hu, 1996*; *Lord & Russell, 2002*). The results of this study are also in agreement with those of *Baghdady et al., 2014*, who reported that the foliar spraying of boron on Valencia orange trees at the full bloom stage significantly increased the number of fruits per tree and the fruit yield (kg/tree) at harvest compared to other treatments and the control. *Gurjar, Kaushik & Baraily (2015)* found that spraying boron, in the form of boric acid, increased the yield of kinnow mandarin.

Calcium (Ca) is involved in fruit drop and pollen tube growth (*Malho & Trewavas, 1996*; *Arrington & DeVetter, 2017*; *Abdel-Sattar, Haikal & Hammad, 2020*), so calcium concentrations in the stigma and style influence the efficacy of pollen germination and tube elongation. Calcium deficiency reduces the strength of the middle lamella and makes cells more prone to shearing, leading to fruit drop (*Arrington & DeVetter, 2017*). Ca is an important signaler of auxins, which in turn reduces the abscission process of leaves, flowers, and fruits (*Sawicki et al., 2015*). Calcium's primary roles are to prevent an abscission zone from forming between fruit pedicles and fruit bearing branches, and to regulate enzyme activity and photosynthesis (*Tony & John, 1994*). Exogenous calcium thus stabilizes the plant cell wall and protects it against cell wall disintegrating enzymes, which significantly influences fruit set and fruit retention (*White & Broadly, 2003*). A logical explanation for the decrease in fruit drop observed with calcium sprays may be calcium's ability to improve cellulose and lignin formation, which are required for building plant structure and for preventing abscission layer formation. Calcium's positive affect on fruit set may be attributed to calcium's ability to improve the efficiency of photosynthesis. TCalcium is also involved in hormone metabolism, stimulating the production of auxins, which are required for fruit set and growth (*Kazemi, 2014*). Calcium improves plant productivity because calcium plays a crucial role in various plant physiological processes including plant growth and development, the structure of the cell wall and cell membranes, cell division, cytoplasmic streaming, and photosynthesis, and it is an obligate intracellular messenger coordinating responses to numerous developmental cues and environmental challenges (*Elmer, Spiers & Wood, 2007*; *Huang et al., 2017*; *Al-Saif et al., 2022b*). *Samaan et al. (2001)* also observed a significant decrease in fruit drop percentage of Washington navel orange trees with CaCl$_2$ treatments, with the most significant effect observed in trees sprayed with 8 g/L CaCl$_2$ solution 3–4 times prior to harvesting. Similarly, *Aly et al. (2015)* reported that spraying Washington navel orange trees with calcium chloride was very effective in improving yield. *Hikal, Ibrahim & Abdelaziz (2017)* found that most boron treatments and the highest concentrations of calcium applications led to a significant decrease in the June fruit drop percentage of Washington navel oranges.

## Fruit physico-chemical characteristics

Seaweed extract application enhanced most physical and chemical fruit quality parameters in this study. This result is in line with those from *Salama (2015)*, who reported that 2% algae extract induced a high positive effect on the fruit quality of Valencia orange trees, and the results of *El-Badawy (2017)*, who showed that algae extract foliar spray greatly improved Valencia orange fruit weight, volume, length, and diameter, as well as the fruit juice percentage, fruit juice TSS percentage, vitamin C, total sugar content, and TTS/acid ratio. In addition, *Mosa et al. (2023)* showed that 0.3% or 0.4% SW application to Anna cultivar apple trees significantly enhanced fruit weight, size, length, and diameter, as well as total soluble solids percentage, total sugar content, reducing sugar content, and non-reducing sugar content compared to untreated trees. *Al-Musawi (2018)* reported that applying SW at 0, 1, 2, and 3% on sour orange trees improved the length, width, and size of fruit, fruit fresh weight, peel thickness, fruit moisture, juice percentage, peel moisture,

ascorbic acid, and total soluble solids, and reduced juice acidity compared with the control treatment. *Yang et al. (2023)* found the foliar application of SE improved the levels of soluble solids, Vitamin C, free amino acids, sugars, and mineral elements in the fruits. Likewise, *Ayub et al. (2019)* reported that SW applied to 'Gala' apple at concentrations ranging from 0.1% to 0.6% resulted in higher fruit set percentage, total fruit numbers, fruit weight, and fruit length compared to the control, with the 0.3% concentration yielding the best results. *Alebidi et al. (2021)* reported that spraying algae on Barhee date palm enhanced the physical and chemical properties of the fruit and yielded the highest values of fruit length, fruit diameter, fruit shape index, flesh weight, seed weight, total soluble solids (TSS), acidity, TSS/acid ratio, tannins, reducing sugars, nonreducing sugars, and total sugars.

Seaweed extract improves fruit quality by affecting major and minor nutrients, amino acids, vitamins, and growth regulators, which affect cellular metabolism, cell division, and cell elongation during the early stages of fruit growth in treated plants, leading to enhanced growth, crop yield, and fruit quality (*Khan et al., 2009*; *Du Jardin, 2015*; *Ali, Ramsubhag & Jayaraman, 2021*; *Rana et al., 2023*; *Yang et al., 2023*). The lower acidity in fruits observed with seaweed application might be due to increased sugar buildup, improved sugar delivery into fruit tissues, the conversion of organic acids to sugars (*Soppelsa et al., 2018*), or the fast acid consumption of organic acid in respiration (*Rana et al., 2023*). The improvements in the physical and chemical properties of Valencia orange fruits observed in this study with seaweed application are in harmony with the reports of *Ahmed et al. (2013)* on Valencia orange; *Gamal (2013)* on Washington Navel orange; *El-Sharony, El-Gioushy & Amin (2015)* on Fagri Kalan mango; *Salama (2015)* on Valencia orange; *Hikal, Ibrahim & Abdelaziz (2017)*; on Valencia and Washington Navel orange; and *Merwad et al. (2019)* on Barhee date palms.

The results of this study indicate that spraying Valencia orange trees with calcium chloride and boric acid has a positive effect on properties of fruit quality. The increase in fruit length, diameter, and volume observed in this study after calcium chloride and boric acid treatments, either alone or combined, may be because these sprayed mineral elements play an indirect role in hormonal metabolism, cell division, and cell expansion (*Rani & Brahmachari, 2001*; *Ojha, 2010*). Similarly, *Dutta & Banik (2005)* and *Yadav, Singh & Yadav (2013)*, reported that fruit length and volume increases may be related to enhancements in the internal physiology of developing fruit in terms of correct delivery of water, mineral nutrients, and other compounds required for normal fruit growth and development. Fruit volume, juice volume, average fruit weight, and average fruit pulp weight increases might also be because of higher accumulations of food material due to increased enzymatic activity and endogenous growth hormone contents, which increase the rate of fruit growth. Agrochemicals enhance cell division and elongation, strengthen the middle lamella and cell walls of the fruit (*Tripathi & Shukla, 2009*), synthesize metabolites, and accelerate the mobilization of photoassimilates and minerals from different parts of the plant towards developing fruits, creating a source-sink relationship (*Tamboli et al., 2015*). Moreover, the increases observed in Valencia orange fruit sizes and fruit fresh weights after B and Ca spraying in two seasons may be attributed to B and Ca's

physiological roles in cell elongation and carbohydrate transport to reproductive tissues such as flowers and fruits and the actuation of the sugar and water mobilization in the fruits (*Lakshmipathi et al., 2015*), leading to increased dry matter accumulation within the fruit (*Bhatt et al., 2012*), increasing fruit weight. Increases in juice volume may be directly tied to increases in fruit weight and volume seen with B and Ca application or due to higher levels of food material accumulation from increased enzymatic activity and the strengthening of the middle lamella and cell walls of the fruit (*Tripathi & Shukla, 2009*).

In terms of fruit chemical properties, boron and calcium sprays increased total soluble solids, total sugars, reducing sugars, and non-reducing sugars, likely due to boron and calcium's involvement in enhancing vegetative developing activities, increasing the absorption of nutrients (*Al-Rawi et al., 2012*). Ca and B also play a role in photosynthetic efficiency, boosting produced minerals in the leaves and growing fruit (*Bhatt et al., 2012*; *Sheikh et al., 2021*). Exogenous Ca application improves photosynthesis, the accumulation of mineral elements in leaves, the growth rates of plant leaves, and significantly improves the fruit quality of some fruit trees (*Wang et al., 2022*). Calcium and boron treatments might improve sugar components through calcium and boron's active involvement in the photosynthesis of metabolites and rapid translocation of sugars, improving sugar metabolism and translocation of assimilates from other parts of the plants to developing fruits (*Singh et al., 2018*). The increased TSS seen with the application of micronutrients may be attributed to the quick metabolic transformations of polysaccharides and pectin into soluble compounds and the rapid translocation of these components from leaves to the developing fruits due to improved source-sink relationships (*Tamboli et al., 2015*). Additionally, the greater total soluble solids and TSS/acidity ratio seen with calcium and boron treatment could be due to the efficient translocation of photosynthates to the fruit *via* calcium and boron modulation (*Meena et al., 2017*). The decrease in titratable acidity may be related to increased nucleic acid synthesis as a result of increased availability of plant metabolites, as documented by *Ullah et al. (2012)* in Kinnow mandarin. The increased levels of vitamin C seen in Valencia orange juice fruits due to calcium and boron application may be attributable to increased nucleic acid synthesis from a maximum availability of Valencia orange tree metabolism (*Sajid et al., 2012*). The results of this study highlighted the significant role of $CaCl_2$ treatments on fruit quality properties. Similarly, *Samaan et al. (2001)* showed that pre-harvest $CaCl_2$ treatments significantly increased TSS, TSS/acid ratio, and vitamin C content, and decreased juice acidity in both Navel and Succary oranges compared with untreated trees. Similar results were found by *Baghdady et al. (2014)* who indicated that spraying Valencia orange trees with 300 ppm chelated calcium and boric acid significantly increased fruit quality compared to control trees. In addition, *Wang et al. (2022)* showed that spraying calcium fertilizer on apple trees increased the sugar content, total soluble solid content, and vitamin C content, and decreased the acid content of apple fruits.

## CONCLUSIONS

Based on the results of this study, spraying Valencia orange trees with 10 g/L of seaweed at flower bud differentiation and after 21 days the first treatment followed by $CaCl_2$ at 1g/L +

$H_3BO_3$ at 0.5 g/L at full bloom (80% flowering) and after 21 days the first treatment is recommended to minimize nutrient loss and maximize nutrient use efficiency. This treatment also generated high density of leafy inflorescences and leafless inflorescences, improved fruit set percentage, increased the final yield, and enhanced the physico-chemical properties of the fruit. The foliar application of small doses of SW in winter and Ca and B during the dynamic growth stages of flowers can help farmers significantly increase yields, and improve productivity and fruit quality of Valencia orange trees.

### Funding

This research was funded by the Researchers Supporting Project (number: RSPD2024R707), King Saud University, Riyadh, Saudi Arabia. The funders had no role in study design, data collection and analysis, decision to publish, or preparation of the manuscript.

### Grant Disclosures

The following grant information was disclosed by the authors:
Researchers Supporting Project: RSPD2024R707.
King Saud University, Riyadh, Saudi Arabia.

### Competing Interests

The authors declare that they have no known competing financial interests or personal relationships including financial, non-financial, professional, or personal relationships, including serving as an Academic Editor for Peer that could have appeared to influence the work reported in this article.

### Author Contributions

- Abdullah Alebidi performed the experiments, analyzed the data, authored or reviewed drafts of the article, and approved the final draft.
- Mahmoud Abdel-Sattar conceived and designed the experiments, performed the experiments, analyzed the data, prepared figures and/or tables, authored or reviewed drafts of the article, funding acquisition, and approved the final draft.

### Data Availability

The raw measurements are available in the Supplemental File.

### Supplemental Information

Supplemental information for this article can be found online at http://dx.doi.org/10.7717/peerj.17378#supplemental-information.

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
