# Peer review of "Synergistic effect of seaweed extract and boric acid and/or calcium chloride on productivity and physico-chemical properties of Valencia orange"

_PeerJ, doi:10.7717/peerj.17378_

## Round 0.1 · original submission · Major Revisions

Dear Dr. Abdel-Sattar,

Thank you for your submission to PeerJ.

Based on review reports, I am of the opinion that your article - Synergistic effect of Seaweed Extract and Boric acid and/or Calcium chloride on Productivity and Physico-Chemical Properties of Valencia orange - requires a number of Major Revisions.

You are therefore advised to carefully go through the comments and suggestions raised by the reviewers, and appropriately modify the manuscript to further improve its quality. Specifically, utmost care is needed while improving the Materials and Methods sections including experimental design. Moreover, additional efforts are needed to ensure that all the section of the paper are up to the mark from a writing stand point. For instance, one of the reviewers comments that 'Introduction is very casually written, there is lack of connectivity between the statements'. You are therefore expected to carefully consider each and every comment, and make the necessary changes.

It is important to mention that your revised manuscript will be evaluated again to ensure that the revised version is in sync with reviewers' queries and suggestions.

Hope to receive the revised manuscript in due course of time.

**Language Note:** The review process has identified that the English language must be improved. PeerJ can provide language editing services - please contact us at copyediting@peerj.com for pricing (be sure to provide your manuscript number and title). Alternatively, you should make your own arrangements to improve the language quality and provide details in your response letter. – PeerJ Staff

Reviewer 1 ·

Basic reporting

The manuscript fits well into the scope of the journal, concentrating on improving the yield and quality of orange fruit through micronutrient supplementation.
It's written nicely, however many comments are given for further improvement.

Experimental design

Its well designed

Validity of the findings

Studied for two years

Additional comments

Mentioned the separate word file

Annotated reviews are not available for download in order to protect the identity of reviewers who chose to remain anonymous.

Reviewer 2 ·

Basic reporting

• English required extensive corrections in terms of language, connectivity and typographical mistakes.
• background of research is not sufficient, authours need to refer more work done on sea weed extract and fruit crops and they should report in introduction part
• table and figure are relevant but required modification of tables such as combining the tables to avoid too many tables
• Line 19-20; x should be replaces with ×
• Significant and crucial results needs to be mentioned in abstract such as how much fruit yield increases. What is the fruit size, quality compared to control etc.
Introduction
• Line 38; particu-larly should be corrected as particularly
• Line 50: when you say world names of the other countries should be mentioned not only Riyadh region, Saudi Arabia or else change the statement particular to Saudi Arabia.
• Line 52: low action exchange may be corrected as Low cation exchange
• In background of research how much fruit loss is there due to poor nutrition, what is the quality reduction, how the present nutrient management practices are carried out what is the problem in present management practices should be given to highlight the need of the present study.
• Line 73: cal-cium and Borne must be corrected as calcium and boron
• There are more typographical and grammatical mistakes are seen in text of the manuscript. Authous needs to carefully check the whole manuscript.
• Introduction is too lengthy and needs to restrict to present problem, importance of seaweed extract, boron and CaCl2 importance in fruit crops grown in calcareous soils of arid and semi arid regions.
• Line 137-138 what nutrients are required for blooming in citrus be specific. how transpiration affect nutrient uptake as Ca and B is not related transpiration at all.
• Previous work on seaweed, Ca and B may be included in the introduction
• In hypothesis it is mentioned that improvement in pollen health and ovule fertilization, but no parameters were studied in this study. And also nutreints may be limiting, what nutrients are limiting, how you are addressing their deficiency, since you used only ca and B along with sea weed extract.
• What is the reason for selecting only one concentration od CaCl2 and B in the study.
• Mention what the leaf nutrient status after and before spraying to understand which nutrients was deficit and what happened after the spray.

Experimental design

• Original research falls in the aims and scope of journal
• Research questions are needs to defined properly.
• Methods need more improvement
• Climatic paramaters during the study needs to be given
• Line 185-186: Al-Dosary et al., 2022 is for mineral content or chlorophyll place the citation at proper position.
• Line 194: Nitrogen can’t be estimated by colorimetry. Please confirm and write the correct method followed
• Equation 1 and 2 may rechecked it is mentioned that fruit retention in both equations.
• Equation 1 looks wrong and give appropriate reference
• Line 214-215 total tree yield is not correct, it may be fruit weight per tree x number of trees. The unit of expression may be kg/ha. Recheck the formula
• Line 228: unit of acidity is % or mg citric acid/100 ml. please check and correct it.
• Table 7 what is fruit length means is it circumference or it is different from diameter. Clarify
• Juice volume can be mentioned as ml/fruit
• Figures needs improvement in terms of clarity
• What type of anova is performed, whether DMRT is used or Tukey test is used. How the graphs were plotted etc needs to be mentioned.

Validity of the findings

Pooled analysis may be done considering year as one factor
• How the shape index is calculated. It is not explained in methodology
• Tables may be restructured including main plot and subplot in one table along with pooled analysis for better clarity and readability. It also reduces the number of tables
• Effect of year has be mentioned in both results and discussion
• Lines 261-262, what is water only treatment and control in the present study s both same. If same use uniform reporting either control or only water spray in text.
• Discussion is very poorly written without explaining the mechanism of ca and B in improving the improvement of other nutrients and also how they are helping in increasing the fruit characters.
• Very generalized statements were made in discussion how SWE influences the growth and physiology. Discussion must be specific and mechanism to explain what particularly caused the beneficial effect of SWE by taking into considerations of previous research. It is simply mentioned that ….et al reported similar results.
• It was mentioned in lines 459-461 that ameliorating the rate of photosynthesis and stomatal opening in plants, and enhancing plant efficacy in up-taking mineral elements such as phosphorous and nitrogen that are directly linked to chlorophyll formation in tree leaves. Since in this study no photosynthesis and stomatal conductance is measures it is not possible to say SWE helped Valencia orange.
• No where in discussion or results is mentioned that how much yield enhancement is seen, how much quality is improved. Etc.

Additional comments

Introduction is very casually written, there is lack of connectivity between the statements,. The previous research on Ca, B and SWE on fruit plants might have discussed to highlight the background of research. Introduction part is too lengthy.
In hypothesis it mentioned about improvement on pollen health fertilization, but no study is carried out on these parameters and its just an assumption.
Authous mentioned previous studies to say SWE, Ca and B helps in uptake of other nutrients, physiology in other crops, but mechanism behind the uptake and suitable parameters to prove the improvement in physiology Valencia orange is not recorded. In this context, it is very general to say SWE, Ca and B helped the trees.
Authors only recorded nutrient improvement in leaves, and fruit characters. Reasons or investigation for higher fruit set and lower fruit drop is not done. Without this effect of SWE, Ca and B in improving on these parameters is not provable. Study also needs pooled analysis considering the two years data.
No where in the discussion or results, the values of observations is not mentioned. i.e., how much yield enhancement is seen, how much quality is improved. Etc. It is only mentioned that SWE + ca+ B improved the yield and its parameters except for one or two observation.

Reviewer 3 ·

Basic reporting

1. Article written nicely and technically sound.
2. Literature sufficient, old references may be replaced with recent ones.
3. Data are illustrated with sufficient tables and figures in the article
4. Research work mainly focuses to enhance citrus yield by using sea weed extract and chemicals in combinations, so worth for publication to benefit farming community and other researchers.

Experimental design

Materials and methods described with sufficient information’s. Need to add available soil nutrients status in table 1.

Validity of the findings

The article is informative provides comprehensive insight of application of seaweed extract (SW) combined with calcium chloride and boric acid to improve fruit yield of citrus crop with other physico -chemical properties.

Additional comments

Article may be accepted after incorporating suggested corrections in the manuscript.

---

## Round 0.2 · Minor Revisions

Dear Dr. Abdel-Sattar,

Thank you for your submission to PeerJ.

It is my opinion as the Academic Editor for your article - Synergistic effect of Seaweed Extract and Boric acid and/or Calcium chloride on Productivity and Physico-Chemical Properties of Valencia orange - that it requires a number of Minor Revisions.

You are therefore advised to go through the reviewers' comments, and submit the revised draft in due course.

Reviewer 2 ·

Basic reporting

• clear and unambigous english is used in the manuscript
• authours mentioned sufficient references citing background of research.
• Introduction is sufficient and with proper hypothesis.

Experimental design

Material and methods written well.

Validity of the findings

• Figure 5-7 1st season and 2ns season, in other figures it was mentioned that 2021 and 2022. Maintain uniform legends in all figures
• Provide LSD values in Fig 5-7
• Tables: abbreviations used should be given in foot note of table
• What is leafy inflorescence? Recheck

Additional comments

i congradulate the authours for incorpotationg all the correction suggested and comments made. however few corrections may be made as mentioned.

---

## Round 0.3 · accepted · Accept

Dear Dr. Abdel-Sattar,

Thank you for your submission to PeerJ.

I am writing to inform you that your manuscript - Synergistic Effect of Seaweed Extract and Boric Acid and/or Calcium Chloride on Productivity and Physico-Chemical Properties of Valencia Orange - has been Accepted for publication. Congratulations!

Reviewer 2 ·

Basic reporting

basic reporting is acceptable

Experimental design

methodology is appropriate and well written

Validity of the findings

all corrections are incorporated as suggested